# Global Research Activities on Micro(nano)plastic Toxicity to Earthworms

**DOI:** 10.3390/toxics11020112

**Published:** 2023-01-24

**Authors:** Wenwen Gong, Haifeng Li, Jiachen Wang, Jihua Zhou, Haikang Zhao, Xuexia Wang, Han Qu, Anxiang Lu

**Affiliations:** 1Institute of Quality Standard and Testing Technology, BAAFS (Beijing Academy of Agriculture and Forestry Sciences), Beijing 100097, China; 2Institute of Plan Nutrition, Resources and Environment, BAAFS (Beijing Academy of Agriculture and Forestry Sciences), Beijing 100097, China; 3Beijing Agricultural Technology Promotion Station, Beijing 100029, China; 4Key Laboratory of Eco-Environment of Three Gorges Region of Ministry of Education, Chongqing University, Chongqing 400045, China

**Keywords:** micro(nano)plastic toxicity, earthworms, bibliometric analysis, web of science, knowledge map

## Abstract

Micro(nano)plastics are emerging contaminants that have been shown to cause various ecotoxicological effects on soil biota. Earthworms, as engineers of the ecosystem, play a fundamental role in soil ecosystem processes and have been used as model species in ecotoxicological studies. Research that evaluates micro(nano)plastic toxicity to earthworms has increased greatly over the last decade; however, only few studies have been conducted to highlight the current knowledge and evolving trends of this topic. This study aims to visualize the research status and knowledge structure of the relevant literature. Bibliometrics and visualization analyses were conducted using co-citations, cooperation networks and cluster analysis. The results showed that micro(nano)plastic toxicity to earthworms is an emerging and increasingly popular topic, with 78 articles published from 2013 to 2022, the majority of which were published in the last two years. The most prolific publications and journals involved in this topic were also identified. In addition, the diversity of cooperative relationships among different countries and institutions confirmed the evolution of this research field, in which China contributed substantially. The high-frequency keywords were then determined using co-occurrence analysis, and were identified as exposure, bioaccumulation, soil, pollution, toxicity, oxidative stress, heavy metal, microplastic, *Eisenia foetida* and community. Moreover, a total of eight clusters were obtained based on topic knowledge clustering, and these included the following themes: plastic pollution, ingestion, combined effects and the biological endpoints of earthworms and toxic mechanisms. This study provides an overview and knowledge structure of micro(nano)plastic toxicity to earthworms so that future researchers can identify their research topics and potential collaborators.

## 1. Introduction

Microplastics (size < 5 mm) and nanoplastics (<100 nm) are emerging contaminants that have attracted worldwide attention due to their ubiquitous and pervasive distribution in nature, as well as their negative effects on ecosystems [1]. Initially, the occurrence, origin and fate of micro(nano)plastics were extensively studied in marine sediments, beaches, shorelines, the surface and deep waters of oceans [2,3], as well as in freshwater systems [4,5]. Subsequent research then suggested that micro(nano)plastics could also enter and accumulate abundantly in terrestrial ecosystems, with soil being one of the largest sinks/reservoirs of this pollutant [6,7,8].

Soil plays important roles in ecosystem functions, such as maintaining soil microclimates and biodiversity, sustaining water supplies and providing food [9]. However, increasing (micro)plastic pollution in soils is now threatening these processes, including the associated plant and faunal communities [10,11,12,13]. As one of the most typical saprozoic soil invertebrates, earthworms play a fundamental role in ecosystem activities, and accordingly, they are regarded as ecosystem engineers [14]. Moreover, since they are relatively easy to maintain under controlled laboratory conditions and have a short generation time, earthworms (*Eisenia fetida* and *Eisenia andrei*) have been used as model species in ecotoxicological studies since the 1980s [15,16]. Given the current apprehension regarding the negative effects of micro(nano)plastics on the terrestrial ecosystem, much attention has been recently given to their toxicity to earthworms, especially in terms of DNA damage, metabolic disorders and other adverse impacts to the species’ growth, survival, reproduction, avoidance behavior and oxidative responses as induced by micro(nano)plastics [17,18,19,20,21]. Although research on the toxicity of micro(nano)plastic to earthworms has made significant progress over the last decade, only few studies have been conducted to highlight the current knowledge and evolving trends of this topic. With an increasing number of articles related to this topic, a retrospective systematic analysis of published literature on this topic is, therefore, warranted.

Bibliometrics is an effective tool for objectively understanding emerging trends and knowledge structures in a research field by quantitatively analyzing patterns in the scientific literature [22]. In this study, a bibliometrics and visualization analysis in the field of micro(nano)plastic toxicity to earthworms was performed, through a series of defined indicators to visualize, analyze and plot this field. Finally, this work briefly discussed the dynamics and future perspectives while providing directions for the further development of the field.

## 2. Data Sources and Analysis Methods

### 2.1. Search Strategy and Data Collection

The Web of Science (WoS) is a comprehensive and influential citation database with mapping techniques and scientific knowledge, which is considered to be an effective source of data acquisition for scientific econometric analysis. Data from the Web of Science Core Collection are collected and analyzed for their representativeness and accessibility. For this study, “Topic Search = earthworm* AND (microplastic* OR nanoplastic*)” were used as the search terms. The search was then performed through the ‘Topic Field’, which considers the “Title,” “Abstract” and “Keywords Plus^®^” of a record limited in English. Since papers and reviews have complete research ideas and reliable data [23], the study was, therefore, limited to these only, with the last search conducted on 5 November 2022. All titles and abstracts were carefully screened, and papers not related to the topic were excluded (Figure 1). Finally, a total of 78 records, including 66 research articles and 12 reviews were obtained for the bibliometric analysis.

### 2.2. Analysis Methods

Data with the following information, namely “Web of Science category, publication year, country/region, highly cited papers” were obtained using the “Analyze Results” feature of the WoS. The origin 9.0 software and Microsoft Excel 2016 were then used for statistical analysis of literature types and publication trends, major published journals, contributions of different countries and scientific research institutions to the publications and citation frequency of articles. In addition, CiteSpace conducts similarity measurements of knowledge units based on data standardization of a set theory, thereby focusing on finding key points in the development of a field, especially research turning points and key points. This approach facilitates an understanding of the development and trend of a field [24]. For this study, the 6.1.R3 version of CiteSpace software (Chen Meichao, Drexel University) was used for bibliometric analysis (co-occurrence, co-citation, hotspots, and cluster analysis). Data processing took place between 2013 and 2023, and the time slice was one year, since the earliest document published in 2013.

## 3. Quantitative Analysis of Basic Information

### 3.1. Annual Publication Trend

A statistical analysis was made on 78 articles related to micro(nano)plastic toxicity to earthworms from the Core Collection database of WoS (Figure 2). Although microplastic pollution has raised increasing serious environmental and public concerns worldwide science 2004, formally introduced by Tompson et al. [25], the earliest publication discussing micro(nano)plastic toxicity in earthworms is less than a decade old, and, in fact, most articles related to this topic have only been published since 2019 (Figure 2), thus indicating that this is an emerging area of interest. The first article in the WoS database was surprisingly detected in 2013, which was related to the earthworm’s exposure (*Eisenia fetida*) to artificial soil amended with PBDE-containing polyurethane foam (PUF) microparticles [26]. Afterwards, Huerta Lwanga et al. [19] provided the first experimental evidence of the effects of polyethylene microplastic on earthworm *Lumbricus terrestris* (Oligochaeta, Lumbricidae). Starting in 2021, the annual publication volume has increased substantially, possibly due to the global awareness of microplastic pollution in the terrestrial ecosystem. Yet, to date, research on micro(nano)plastic toxicity to earthworms remains very limited. Nevertheless, the increasing number of publications and citations in this field (average number of citations per publication up to 43.1) over the past seven years indicates that research on this topic has attracted extensive interest, thereby highlighting the concern of the worldwide scientific community.

### 3.2. Analysis of Published Journals

As part of the bibliometric analysis, journal analysis provides readers with a better understanding of the variety of publications for a particular topic. In this context, citations and impact factors (IF) are the most common bibliometric evaluation variables. The collected articles that fell within the scope of this study were published in 29 journals, with the top five shown in Table 1. More specifically, Science of the Total Environment ranked first with 19 publications (24.4% of the total publications), followed by Environmental Pollution with 11 publications (14.1%) and the Journal of Hazardous Materials with 8 publications (10.3%). While Environmental Pollution had the second highest number of publications, it also had the highest citation number (1276), indicating that it made important contributions to research on micro(nano)plastic toxicity to earthworms. These journals were more frequently cited by researchers in this field, hence demonstrating their considerable influence. For instance, on average, a single article of Environmental Science & Technology had been cited as high as 156 times, thus reflecting the high quality of the articles published in this journal and that it is favored by the majority of scientific research workers.

### 3.3. Cooperative Relationship Network

The cooperative relationship network was conducted to help identify the key countries and research institutions that published a large number of papers, thereby making a significant impact in the field (Figure 3 and Figure 4). Through the knowledge map analysis, 96 institutions from 31 countries or regions were engaged in the field of micro(nano)plastic toxicity to earthworms. China, in particular, had published a total of 40 articles, accounting for around 51.3% of all publications, followed by the United States, Australia and South Korea with six publications each. In terms of centrality, China had a number of research collaborations with different countries such as the United States, Netherlands, Portugal and Australia, etc., thus forming key nodes with the highest centrality of 0.87, indicating the strongest influence on the relationship within the cooperation networks, followed by the Netherlands with a centrality of 0.43. Through the network map, the centrality of the United States, Australia and South Korea was low (<0.01), although their publication numbers ranked second (each of them with six publications). These figures indicated that scientists in these countries worked more independently. The Chinese Academy of Sciences, the University of Chinese Academy of Sciences and China Agricultural University were the three main institutions that issued the highest number of publications with 12, 9 and 7 papers, respectively. Figure 4 illustrates the cooperation network between the different institutions on this topic. Essentially, two main institutional groups of cooperation were identified. The first cooperative network was led by the Chinese Academy of Agriculture Sciences (centrality of 0.09), the Chinese Academy of Sciences (centrality of 0.06) and China Agricultural University (0.03). This collaborative network covered many research institutions and universities in China and North America. In addition, the University of Aveiro (0.01), the University de Vigo, the University of Porto and some others formed the second cooperative network, which also included a number of research institutions and universities across European countries.

## 4. Document Co-Citation Analysis

Document co-citation analysis not only provides an active way of identifying knowledge domains and intellectual structures, but also reflects the current research directions, as well as hotspots in a particular field. The 10 most cited documents, based on their co-citations, are listed in Table 2. Among the first ten most-cited articles, only one was a review, while the rest were research papers. In terms of the journals that published the most-cited articles, Environmental Pollution ranked first with six publications, followed by Environmental Science & Technology with three publications, while Journal of Hazardous Materials had only one publication. The most cited paper was published by Huerta Lwanga et al. [19] in 2016 where they investigated the survival and fitness of the earthworm *Lumbricus terrestris* after exposure to a high concentration (up to 60% dry weight) of polyethylene microplastics. The results confirmed that polyethylene microplastics could be directly ingested by earthworms. Following this study, other researchers used lower exposure doses which were more comparable to environmentally relevant concentrations to study the effects of microplastics on earthworms [20,27,28,29]. Furthermore, the impact of micro(nano)plastics on the bioaccumulation of other contaminants in earthworms represents another significant research domain [30,31]. In particular, several studies investigated the combined effects of microplastics and heavy metals (e.g., zinc and arsenic) [32,33] or organic chemicals [26,34]. This topic will be further discussed below.

## 5. Analysis of Hotspot Evolution

Generally, keywords can effectively reflect research hotspots which represent topics of wide concern/interest for researchers in a specific field. Figure 5 shows the results for a keyword co-occurrence network which was established with 156 keywords and 809 links. In this case, the modularity was Q = 0.4453, while the mean Sihouette was 0.7078. (Figure 5a). The nodes in the network indicated the corresponding keywords, with the node size being positively associated with the occurrence frequency of the keyword. Additionally, nodes that had a co-occurrence relationship were connected by a colored line [36]. Overall, there was a complex relationship between the keywords, as shown by their intricate connections (Figure 5a). The top ten keywords included: exposure (centrality of 0.05), bioaccumulation (centrality of 0.20), soil (centrality of 0.15), pollution (centrality of 0.04), toxicity (centrality of 0.04), oxidative stress (centrality of 0.04), heavy metal (centrality of 0.09), microplastic (centrality of 0.13), *Eisenia foetida* (centrality of 0.01) and community (centrality of 0.04). These highly frequent keywords, as well as the high score for betweenness centrality, were significant for the development of the field of micro(nano)plastic toxicity to earthworms, and contributed to linking and merging different topics.

With the assistance of CiteSpace software, all extracted keywords were sorted into different clusters based on their co-occurrence relationships to describe the structure of the research field. As shown in Figure 5b and Table 3, there were eight clusters, namely plastic pollution, sorption/desorption, ingestion, heavy metals, gene expression, bioaccumulation, histopathology and gut microbiota. The silhouettes for these eight significant clusters were all between 0.677 and 0.827, hence indicating consistency among cluster members. As some of the clusters had similar themes, they were described together below.

Cluster #0 “plastic pollution” had 29 members and the Silhouette value was very close to 0.7, thus indicating a reliable clustering result. Plastic pollution represents an urgent environmental problem worldwide, with rapid urbanization and a continuous increase in the demand and consumption of plastic products further exacerbating the issue. Each year, an enormous amount of plastic waste enters the environment and breaks down into small pieces as a result of photolytic, mechanical and biological degradation to yield micro(nano)plastics [37]. In recent decades, there has been considerable evidence of the widespread presence of micro(nano)plastics in marine, freshwater and terrestrial ecosystems [3,4,5,8]. In fact, it has been recognized that microplastic pollution in terrestrial ecosystems tends to be worse (4–23-fold larger) than that in oceans [38]. Over the last decade, there has been increasing research on how plastic, especially micro(nano)plastic pollution, affects soil ecosystems (e.g., ecosystem functions), soil microbes, plants and animals [12,39,40,41]. However, the study of these pollutants’ ecotoxicity has been limited due to the complexity and heterogeneity of soil systems [42]. In this context, some researchers stated that ecological interactions between micro(nano)plastic and soil communities may play a key role in determining their effects on terrestrial ecosystems [9]. Therefore, identifying the impacts and mechanisms of micro(nano)plastic ecotoxicity (especially on whole communities) may help to better assess the associated risks, even though much work remains to be done in this regard.

Cluster #2 “ingestion” had 27 members and a high Silhouette value (larger than 0.7). It has been shown that micro(nano)plastics can be ingested by earthworms due to their small sizes [30,43]. For instance, Wang et al. [34] used Nile red fluorescence imaging and clearly observed the ingestion of polyethylene (PE, ≤300 μm) and polystyrene (PS, ≤250 μm) particles by the earthworm *Eisenia fetida*. Some researchers, in fact, believed that earthworms were more inclined to choose polymers containing biodegradable components, e.g., poly(ethylene terephthalate) (PET) and poly(lactic acid) (PLA), presumably because of the odor of the polymer monomers [44]. There were, however, some researchers who argued that earthworms had no component-specific preference for the ingestion of plastics [45,46]. These contrasting findings could be due to a number of factors such as the soil properties, particle sizes and exposure time, and thus, further studies, as well as extended experiments, would be required to resolve the above contrasting views. It has been reported that earthworms ingested plastic debris and actively contributed to break them into smaller particles [44]. Indeed, submicron and nanocron plastic particles were found in earthworm casts, and this may contribute to their diffusion into deeper soils and even to groundwaters where they can potentially impact other relevant components of the terrestrial ecosystem. Additionally, it has been reported that microplastics can also be indirectly taken by predators (e.g., chicken) after being ingested by earthworms, thus leading to trophic transfer [47,48].

Cluster #1, #3 and #5 were focused on the combined effects of micro(nano)plastic and other co-occurring contaminants in earthworms. Numerous studies have demonstrated that the small particle size and large specific surface area of micro(nano)plastics make them capable of absorbing ambient pollutants, such as heavy metals and hydrophobic organic chemicals (HOCs) [49,50,51]. Their combined toxicity has, therefore, been the focus of many scholars. In this context, some evidence has shown that micro(nano)plastics could increase the bioaccumulation of heavy metals Cu, Ni [52] and Cd [53], as well as HOCs dufulin [54], pyrene [55], PFOS and PFOA [56] in earthworms by enhancing their bioavailability. Conversely, other researchers suggested that microplastics, at amendment rates of ≥1% d.w., reduced the tissue concentrations of polycyclic aromatic hydrocarbons (PAHs) and polychlorinated biphenyl (PCB) in earthworms [34], and this could potentially be related to the fugacity gradient of chemicals between microplastics and soils [57]. In general, combined exposure can induce greater oxidative damage to earthworms compared with exposure to micro(nano)plastics or other pollutants alone, and, thus, negative effects on growth, behavior and the immune system of earthworms can be heightened [20,53,54]. In addition to the increased bioaccumulation of contaminants in earthworms, an increase in intestinal permeability and changes in gut bacterial communities induced by micro(nano)plastics are important factors that can potentially enhance oxidative damage upon combined exposure [30,58].

Cluster #1, #3 and #5 were focused on the combined effects of micro(nano)plastic and other co-occurring contaminants in earthworms. Numerous studies have demonstrated that the small particle size and large specific surface area of micro(nano)plastics make them capable of absorbing ambient pollutants, such as heavy metals and hydrophobic organic chemicals (HOCs) [49,50,51]. Their combined toxicity has, therefore, been the focus of many scholars. In this context, some evidence has shown that micro(nano)plastics could increase the bioaccumulation of heavy metals Cu, Ni [52] and Cd [53], as well as HOCs dufulin [54], pyrene [55], PFOS and PFOA [56] in earthworms by enhancing their bioavailability. Conversely, other researchers suggested that microplastics, at amendment rates of ≥1% d.w., reduced the tissue concentrations of polycyclic aromatic hydrocarbons (PAHs) and polychlorinated biphenyl (PCB) in earthworms [34], and this could potentially be related to the fugacity gradient of chemicals between microplastics and soils [57]. In general, combined exposure can induce greater oxidative damage to earthworms compared with exposure to micro(nano)plastics or other pollutants alone, and, thus, negative effects on growth, behavior and the immune system of earthworms can be heightened [20,53,54]. In addition to the increased bioaccumulation of contaminants in earthworms, an increase in intestinal permeability and changes in gut bacterial communities induced by micro(nano)plastics are important factors that can potentially enhance oxidative damage upon combined exposure [30,58].

Cluster #4, #6 and #7 were involved in multiple biological endpoints and toxic mechanisms in the growth, reproduction, survival, lifespan, ingestion rate and avoidance behavior of earthworms exposed to micro(nano)plastics. Micro(nano)plastics have been known to adversely affect earthworms, with direct evidence coming from exposure experiments [18,19,27] which highlighted the adverse effects on the animals’ mortality, growth rate, ingestion rate and accumulation. At the same time, these effects were intimately related to their types, sizes, exposed concentrations and time, amongst others, as confirmed by a recent meta-analysis [59]. In contrast to the growth and survival indicators, other indicators at the cell, tissue or gene level were more sensitive to micro(nano)plastic exposure at the environmentally relevant concentrations. For example, Rodriguez-Seijo et al. [20] tested the effects of polyethylene microplastics on the survival and fitness on earthworms (*Eisenia andrei Bouché*). They reported that a 28-day exposure to microplastics did not significantly affect the animals’ survival and growth, although damages to the gut (e.g., gut inflammation) and immune responses to microplastics were clearly observed. Similarly, Jiang et al. [27] reported impaired intestinal cells upon exposure to polystyrene microplastics, with oxidative stress and DNA damage also involved. These more sensitive indicators or biomarkers (such as enzymes, genes and metabolites) could help to further explain how earthworms are being affected by micro(nano)plastics, including sub-lethal but chronic effects.

## 6. Summary and Prospects

Micro(nano)plastic research is a rapidly evolving domain. Despite recognizing that soils have the potential to retain and accumulate micro(nano)plastics, much work remains in the field of micro(nano)plastic toxicity to soil biota (e.g., earthworms) which are involved in key soil functions and related ecosystem processes. In this study, global research activities of micro(nano)plastic toxicity to earthworms were collected, and the current research performance and knowledge structure on this topic were performed by CiteSpace-based visualization analysis. Results showed the increasing trend on micro(nano)plastic toxicity in earthworms, as well as active collaborations between multiple countries and institutions for research on this topic. Research hotspots related to themes of plastic pollution, ingestion, combined effects, the biological endpoints and toxic mechanisms in earthworms exposed to micro(nano)plastics were also identified. In general, it is essential to thoroughly investigate the behavior and/or physiological response of earthworms upon exposure to micro(nano)plastics in order to better evaluate their impacts on the health of the soil ecosystem.

More studies are, therefore, needed to fully understand the interactions between micro(nano)plastics and earthworms under long-term realistic field conditions, as well as at realistic exposure concentrations, since most of the available studies focus on high-dose short-term exposures under (semi-)controlled laboratory conditions. Besides, most existing research used *Eisenia fetida* or *Eisenia andrei* as the model species, but it should be acknowledged that these species are rarely found in agricultural soils and are often less sensitive to pollutants compared with other earthworm species found in mineral soils, e.g., *Aporrectodea caliginosa* [60]. To better assess the toxic effects of micro(nano)plastics on non-target soil animals (including earthworms), there is a need to perform a posteriori tests using relevant species. Lastly, soil animals co-inhabit and interact with soil microbes and plant species. Future research is needed to address the impacts and ecological risks of micro(nano)plastics from a holistic viewpoint.

## Figures and Tables

**Figure 1 toxics-11-00112-f001:**
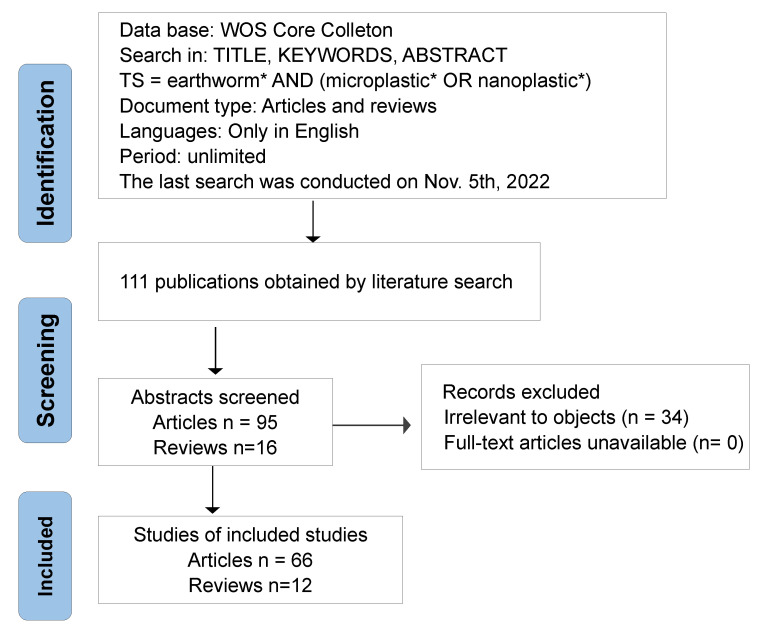
Search strategy implemented to obtain relevant publications for the bibliometric analysis (n denotes the number of publications at the different filtering stages).

**Figure 2 toxics-11-00112-f002:**
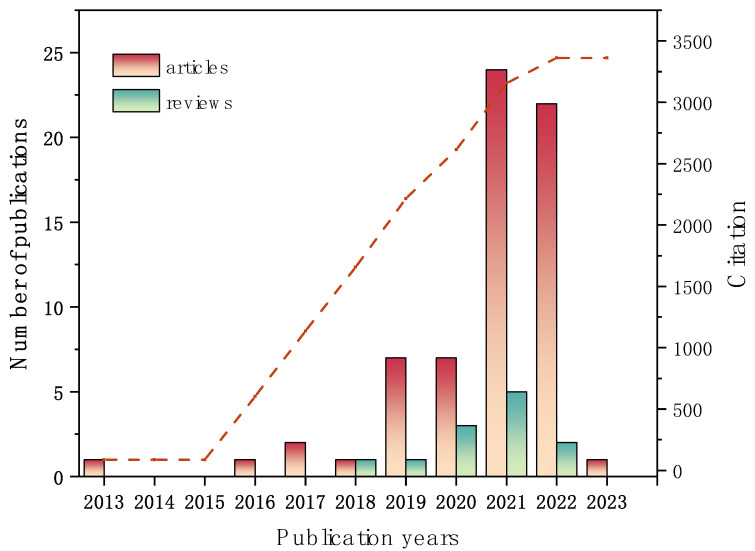
Annual number of publications and citation trends (red dashed line) for research on micro(nano)plastic toxicity to earthworms.

**Figure 3 toxics-11-00112-f003:**
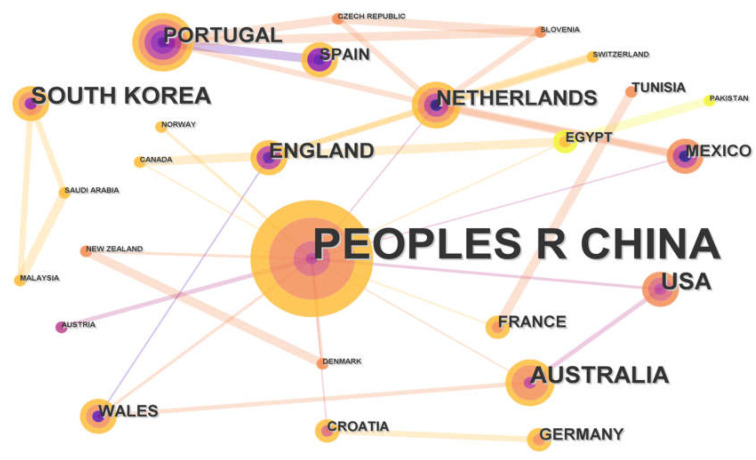
Cooperation networks for countries in the field of micro(nano)plastic toxicity to earthworms.

**Figure 4 toxics-11-00112-f004:**
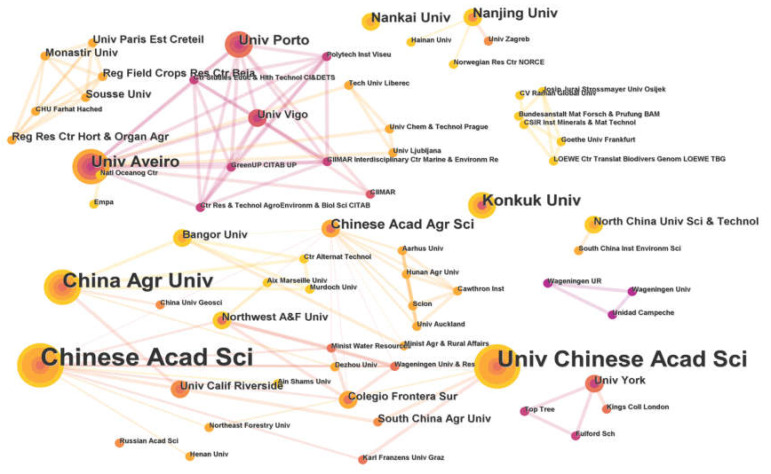
Institution cooperation network in the field of micro(nano)plastic toxicity to earthworms.

**Figure 5 toxics-11-00112-f005:**
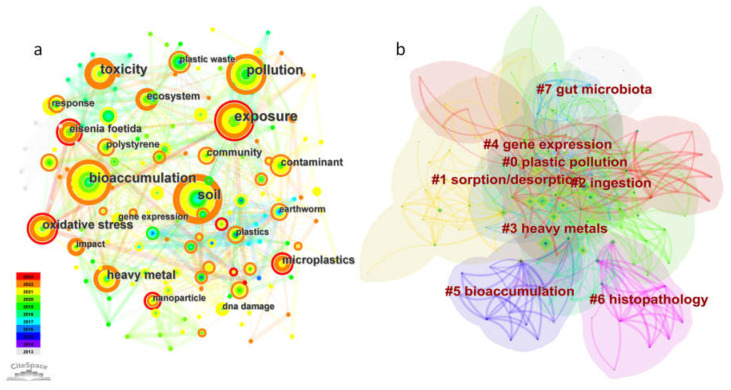
Co-occurrence keyword network (**a**) and co-occurrence clustering keyword network (**b**) in the field of micro(nano)plastic toxicity to earthworms.

**Table 1 toxics-11-00112-t001:** Top five journals in the field of micro(nano)plastic toxicity to earthworms.

Journal	Number	Percentage (%)	IF (2022)	Citation	Citation/n
Science of the Total Environment	19	24.4	10.753	229	12.1
Environmental Pollution	11	14.1	9.988	1276	116
Journal of Hazardous Materials	8	10.3	14.223	264	33.0
Environmental Science & Technology	6	7.7	11.357	936	156
Chemosphere	5	6.4	8.943	74	14.8

**Table 2 toxics-11-00112-t002:** Top 10 highest cited papers in terms of citation frequency in the field of micro(nano)plastic toxicity in earthworms.

Rank	Title	Year	Source	Total Citation	Paper Types	References
1	Microplastics in the terrestrial ecosystem: implications for *Lumbricus terrestris* (Oligochaeta, Lumbricidae)	2016	Environmental Science & Technology	516	Article	[19]
2	Current research trends on plastic pollution and ecological impacts on the soil ecosystem: A review	2018	Environmental Pollution	443	Review	[35]
3	Plastic bag derived-microplastics as a vector for metal exposure in terrestrial invertebrates	2017	Environmental Science & Technology	289	Article	[32]
4	Histopathological and molecular effects of microplastics in *Eisenia andrei* Bouche	2017	Environmental Pollution	245	Article	[20]
5	Negligible effects of microplastics on animal fitness and HOC bioaccumulation in earthworm *Eisenia fetida* in soil	2019	Environmental Pollution	122	Article	[34]
6	Toxicological effects of polystyrene microplastics on earthworm (*Eisenia fetida*)	2020	Environmental pollution	97	Article	[27]
7	Microplastic digestion generates fragmented nanoplastics in soils and damages earthworm spermatogenesis and coelomocyte viability	2021	Journal of Hazardous Materials	90	Article	[29]
8	Exposure to microplastics lowers arsenic accumulation and alters gut bacterial communities of earthworm Metaphire californica	2019	Environmental Pollution	90	Article	[33]
9	Microplastics in municipal mixed-waste organic outputs induce minimal short to long-term toxicity in key terrestrial biota	2019	Environmental Pollution	88	Article	[28]
10	Polybrominated diphenyl Ether (PBDE) accumulation by earthworms (*Eisenia fetida*) exposed to biosolids-, polyurethane foam microparticle-, and penta-BDE-amended soils	2013	Environmental Science & Technology	87	Article	[26]

**Table 3 toxics-11-00112-t003:** Knowledge clusters in the field of micro(nano)plastic toxicity to earthworms based on keyword co-occurrence.

Cluster ID	Cluster Label	Size	Silhouette	Top Terms
#0	plastic pollution	29	0.696	plastic pollution; ecotoxicity; ecosystem engineers; macroplastic
#1	sorption/desorption	27	0.677	sorption; desorption; hydrophobic organic chemical; pollutant; contaminant
#2	ingestion	27	0.725	ingestion; earthworm response; plastics; size; ecosystem health
#3	heavy metals	19	0.713	heavy metals; oxidative stress; combined exposure; *Eisenia fetida*; distribution pattern
#4	gene expression	19	0.701	gene expression; antioxidant enzymes; earthworm; histology; glyphosate transport
#5	bioaccumulation	13	0.730	bioaccumulation; geophagous earthworm; flame retardant; consumer product
#6	histopathology	13	0.816	histopathology; lactate dehydrogenase; muti-omics; catalase
#7	gut microbiota	8	0.827	gut microbiota; microplastics; reactivity; physiological systems

## Data Availability

The authors confirm that the data supporting the findings of this study are available within the article.

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
