# Peer review of "Global Research Activities on Micro(nano)plastic Toxicity to Earthworms"

_toxics, 2023, doi:10.3390/toxics11020112_

Round 1
Reviewer 1 Report
Accept in current form
Author Response
Thank you for your encouragement.
Reviewer 2 Report
Remarks to the Author:
Wenwen Gonga and colleagues provides an overview of global research activities on micro(nano)plastic toxicity to earthworms,
It is an interesting work research, but there are still some things that need to be revised.
Overall, I would support the publication of this study after minor revision.
1. The full name "Figure" should be used in the full text, such as “Fig 1” should be written as “Figure 1”.
2. Red dashed line should be clarified in figure legend (line 128).
Author Response
We appreciate the reviewer’s helpful comments and general encouragement. We have added more clarifications and edited the manuscript thoroughly.
- The full name "Figure" were used in the manuscript instead of "Fig.".
- According to the comment, red dashed line should be clarified in figure legend.
Reviewer 3 Report
This manuscript deals with ,,Global research activities on micro(nano)plastic toxicity to earthworms,, and with 78 articles published from 2013 to 2022.
The abstract should be factual and should contain background, results and conclusions.
The language of manuscript should be further improved.
This paper may be helpful. https://www.mdpi.com/2073-4360/14/21/4770
Line 53-58, ecotoxicological studies, earthworms: Very essential information and devoted to it in detail brought this important paper which should be mentioned to improve the introduction. https://www.sciencedirect.com/science/article/pii/S030438942200560X
Give a strong reason why this work is new and what it will bring to the scientific community.
Line 76: The Scopus database should also be checked.
Line 105: Why did you choose this time frame?
Line 128: How do you explain that interest in this issue has increased significantly in 2021 and 2022?
Line 146: Environment International might also be worth considering.
Line 312: List the most important future perspectives.
Author Response
We greatly appreciate the constructive comments from the reviewers, which helped us improve our manuscript substantially.
- According to the reviewer’s suggestion, we have revised the Abstract part.
- The overall quality of language in the manuscript has been further improved.
- This paper by Khaldoon et al. (2022) has been cited.
- We think that the paper provided in the comment is irrelevant since it studed the exotoxicological effects of PFAS in soils on earthworms and plants. By compasion, we have cited two classic referrence in the original manuscript.
- As suggested, we added a general sentence and elaborated on the significance of this study for the scientific community at the revised abstract.
- We fully understand the reviewer's concern, more database would make the picture more complete. However, as we clearly mentioned in the manuscript that WoS database is considered to be an effective source of data acquisition for scientific econometric analysis, and numerous studies have used WoS database for bibliometrics analysis.
- The search time was unlimited. The first article to be published was detected in 2013, so the subsequent analysis was between 2013 and 2023.
- As we have mentioned in the original manuscript, "Starting in 2021, the annual publication volume has increased substantially, possibly due to the global awareness of microplastic pollution in the terrestrial ecosystem".
- Table 1 showed the top five journals in the field of micro(nano)plastic toxicity to earthworms, according to the number of publications in those journals. Environment International is a famous journal, but its publication numbers were not renked in the top five.
- To make the section of Summary and prospects clearer and easy to read, we have made some appropriate changes.
Round 2
Reviewer 3 Report
The manuscript can be accepted.